# Global Climate Change and Human Dirofilariasis in Russia

**DOI:** 10.3390/ijerph19053096

**Published:** 2022-03-06

**Authors:** Anatoly V. Kondrashin, Lola F. Morozova, Ekaterina V. Stepanova, Natalia A. Turbabina, Maria S. Maksimova, Aleksandr E. Morozov, Alina S. Anikina, Evgeny N. Morozov

**Affiliations:** 1Martsinovsky Institute of Medical Parasitology, Tropical and Vector-Borne Diseases, Sechenov University, 119435 Moscow, Russia; anakona@mail.ru (A.V.K.); lfmorozova@mail.ru (L.F.M.); stepan-kate83@mail.ru (E.V.S.); n.turbabina@mail.ru (N.A.T.); maksimovmarij@yandex.ru (M.S.M.); a.e.morozov@mail.ru (A.E.M.); alianikina19@gmail.com (A.S.A.); 2Department of Tropical, Parasitic Diseases and Disinfectology, Russian Medical Academy of Continuous Professional Education, 125993 Moscow, Russia

**Keywords:** dirofilaria, climate change, areal, dirofilariasis

## Abstract

Human dirofilariasis is a vector-borne helminth disease caused by two species of *Dirofilaria*: *D. repens* and *D. immitis*. The vectors of the helminth are mosquitoes in the family Culicidae. The definitive hosts of *Dirofilaria* are dogs and, to a lesser extent, cats. Humans are accidental hosts. Dirofilariasis has been reported in the territory of Russia since 1915. Sporadic cases of the disease have been reported occasionally, but the number of cases showed a distinct increasing trend in the late 1980s–early 1990s, when the number of cases reached several hundred in the southern territories of Russia, with geographic coordinates of 43° N–45° N. A comparison of the timing of the global trend of climate warming during the 1990s with the temporal pattern of the incidence of dirofilariasis in the territory of Russia indicated a close association between the two phenomena. At present, the northern range of Dirofilaria includes latitudes higher than 58° in both the European and Asian parts of the country. The phenomenon of climate warming in the territory of Russia has shaped the contemporary epidemiology of the disease. The emerging public health problem of dirofilariasis in Russia warrants the establishment of a comprehensive epidemiological monitoring system.

## 1. Background

Observations over the last 40–50 years on the dynamics of climate warming in the temperate zone of the northern hemisphere have revealed diversity with respect to different territories. This phenomenon relates to the topology of the territory, the proximity of such topology to the seas and oceans, and alterations in circulatory processes [1,2].

It has been noted that the warming of the climate primarily affects the winter and spring months, and the tempo of climate warming steadily increased at the end of the 20th century [1]. The reciprocal reaction of biota to climate change in the northern hemisphere is a shift in borders and isotherms towards the north [2].

Therefore, it is logical to assume that climatic changes might interfere with the dynamics of parasitic diseases in humans, or with their components. For example, in Sweden, warmer winters have resulted in changes in vector ecology, such as that of the European forest tick, *Ixodes ricinus*, affecting certain parasitic diseases in humans. The border of the range of this tick has shifted northwards [3]. In the Taldom district of the Moscow region of Russia, the density of ticks more than doubled in comparison with the second half of the last century. Moreover, the autumn peak reached the same level as or even exceeded the spring peak [4].

It is a well-known historical fact that a short period of warming during the summer months in the 1930s–1940s in the Arkhangelsk (64°32′ N) and Vologda (59°13′ N) regions of Russia resulted in the mass breeding of *Anopheles maculipennis* mosquitoes, which are malaria vectors, and the occurrence of epidemic outbreaks of *Plasmodium vivax* and *Plasmodium falciparum* malaria [5]. Climate warming facilitated the reestablishment of the local transmission of *P. vivax* malaria in the territory of the Moscow region from the late 1990s to the early 2000s [6,7].

In principle, the climate on Earth is permanently subject to change. For the last 4.5 billion years, modulations of the climate, during which it was considerably warmer or considerably colder than it is now, have occurred. However, never before in history has the speed of climate change been as high as it is at present. On average, climate warming in Russia takes place 2.5 times more rapidly than that in the rest of the planet due to the peculiarities of the territory. The main reason for this phenomenon is the preponderance of land surface over water surface, as the former has a different heat capacity than the latter [8,9].

The present epidemiological situation in Russia with respect to human dirofilariasis is a good illustration of the consequences of climate change on various aspects of the disease, particularly for the last 30–40 years [10].

*D. repens* and *D. immitis* are the only helminths transmitted by Aedes, Culex and Anopheles mosquitoes in territories with temperate climates. *D. repens* is the main causative agent of human and animal (dog, cat) filariasis; cases caused by *D. immitis* are very rare in humans. In Russia, dogs are affected equally by both species of *Dirofilaria*. Clinically, human filariasis manifests as pain, oedema and localized erythema or erythema on the trunk, head, hands and legs [11,12]. One striking characteristic of the disease described by many patients is the feeling of the movement (“crawling”) of the worm under the skin [13]. In the absence of efficacious specific treatments, the main intervention is the surgical removal of the worm. At present, the control and prevention of human dirofilariasis involves the treatment of infected animals, which is a difficult operational task [14].

Similar to any vector-borne disease, ambient temperature is the most important factor for the development of Dirofilaria into the infective larval stage (L3) in the mosquito. This can be achieved at the minimum threshold of 14 °C, and a sum of effective temperatures of 130 °C/day (a “sum of effective temperatures” (SET) or “growing degree day” (adopted in several models in the West) is needed for Dirofilaria larvae to reach infectivity; the concept of SET was originally developed and successfully applied in Russia in the control of malaria) [15]. The maximal life expectancy of an infected mosquito is 30 days.

## 2. Impact of Climate Warming on Dirofilariasis in Russia

There are two major dimensions of the impact of climate warming on the local transmission of dirofilariasis in the territory of the Russian Federation:Climate warming facilitated an expansion of the area of dirofilariasis from the original endemic foci in the southern part of the country northwards (from 43°31′ N to 58°59′ N) and eastwards. At present, local transmission is possible in more than 50% of the total administrative entities of the country.Climate warming has strongly shaped the contemporary epidemiology of dirofilariasis in the territory of the Russia.

### 2.1. Dirofilariasis Expansion Area

Until the end of the 20th century, only sporadic cases of dirofilariasis caused by *D. repens* in humans were reported in the territory of the Russian Federation [16,17,18,19].

Indigenous cases of *D. repens* were reported in a number of regions in the European part of the country, as well as in the Asian and Far Eastern parts. In Europe, the cases were confined to territories with geographic coordinates ranging from 43°10′ N (Grozny, Chechen Republic [19] to 48°71′ N (Volgograd) [20]. Indigenous cases in Asia were reported in territories with geographic coordinates of 53°36′ N (Barnaul, Republic of Altai) [21]. Cases of dirofilariasis in the Far East were reported in Vladivostok (43°01′ N) [22,23].

At the same time, sporadic imported cases of dirofilariasis caused by *D. repens* originating in territories in Russia that had local transmission were reported in the northern regions of the European part in Voronezh (51°42′ N) [24] and Nizhny Novgorod (56°32′ N) [25]. Several imported cases of dirofilariasis caused by *D. repens* were also reported in the Asian part of Russia, in the Tomsk region of Western Siberia (56°49′ N) [26]. These cases suggest that the monitoring of the dirofilariasis spread in Asian Russia may not be as good as it is in European Russia. The dynamics of *D. repens* infection in humans in the territory of the Russia are presented in Table 1.

Prior to the disintegration of the USSR (1991–1993), the majority of dirofilariasis cases were detected in the territory of the Russia (62%) and Ukraine (21%). The remainder of the cases (17%) were registered in Kazakhstan (5%), Uzbekistan (4%), Georgia (4%) and Turkmenistan (4%) [16,27,28].

From the 1960s–1995, the northern border of the Dirofilaria range did not cross the latitude of 53° N–54° N. The majority of cases were registered in the Krasnodar region and in the cities in the Volga River Basin, including Astrakhan, Volgograd, and Saratov [4,14,29].

Overall, the territories with reported indigenous cases were within the isotherm contours representative of the month of January, with temperatures from 0° to +16 °C, and the isotherm contours representative of July, with temperatures ranging from +16° to >32 °C, within areas for which the annual cumulative sun radiation fluctuated between 80–160 Kkal/cm [16].

Starting in the mid-1990s, the incidence of *D. repens* infection in men in the Russia demonstrated an almost exponential increase under the influence of climate warming (Figure 1). This trend is quite similar to that seen worldwide (http://www.crue.uea.ac.uk/cru/data/temperuture (accessed on 17 April 2021)).

From 1997–2012, indigenous *D. repens* infection cases were confined largely to the territories of the European part of the Russia, as compared with the Asian and Far Eastern parts, as shown in Figure 2 and Figure 3.

Since 2012, this ratio has changed considerably, as shown in Figure 3.

As of 2018, out of the 83 administrative units in the territory of the Russia (hierarchical order: Republic, Territory, Region, Autonomous district, District), indigenous cases of *D. repens* infection were recorded in 54. Their geographical positions are shown in Table 2.

It appears that the warming of the climate resulted in the expansion of the range of *D. repens* not only northwards (from 45 N to 58 N) but also eastwards. At present, the northern border of the endemic area of dirofilariasis is beyond 58° N in both the European and Asian parts of Russia.

Based on these data, a simplified approach was deployed for the construction of a map indcating the risk of dirofilariasis in the territory of the Russia [13]. This approach takes into account the notion that *Dirofilaria* transmission depends on successful larval development to a mosquito, which, in turn, requires favorable environmental conditions, particularly a certain level of warm temperature.

To construct the map, long-term average air temperature data for July were obtained from the Meteorological Office of Russia for 1937–2016. July is the warmest month of the year, with the highest density of vectors and microfilariae in the blood of infected dogs. An isotherm of +14 °C for July was chosen as the northern border of the potential dirofilariasis endemic area in the territory of Russia (Figure 4).

At the time of map development, the northernmost areas of the occurrence of indigenous cases of dirofilariasis in the European part of Russia were Novgorod (58°26′ N) and Kirov (58°59′ N) [30]. In the Asian part, the northernmost territory was the city of Tyumen (57°15′ N). However, in 2018, indigenous cases of *D. repens* infection were registered in the town of Kolpashevo (58°19′ N) in the Tomsk region [26]. This finding confirmed that the northernmost border of the indigenous transmission of dirofilariasis was beyond 58° N in both the European and Asian parts of the Russia.

Overall, under the influence of climate warming and within only approximately 20 years, the northern border of the Dirofilaria range in the territory of Russia shifted from 43° N to 58° N, and the risk of infection among the population of the Russia doubled.

### 2.2. Impact of Climate Warming on the Epidemiology of Dirofilariasis in the Russia

#### 2.2.1. Prolongation of the Transmission Season

The most noticeable impact of climate warming on dirofilariasis was an increased duration of the season of local transmission [31,32]. A good indicator of the duration of dirofilariasis transmission is the seasonal pattern of clinical manifestations of the disease and the life cycle of the mosquito vector.

Based on the results of observations carried out in the territory of endemic foci in the southern part of the Russia, seasonal variations in the rates of clinical manifestations of dirofilariasis were calculated for Rostov-on-Don city (47°23′ N) [33]. These rates were 28% in summer and 19.0% in autumn.

The results of other investigations showed that if the peak vector density was in the months of July and August, then the infection would most likely occur during the summer period, and the appearance of clinical manifestations would occur in winter or spring of the following year [34]. Empirically, it was established that the incubation period could be 8–10 months or longer [35].

However, recent similar investigations in the Nizhny Novgorod region (56°30′ N) [25] (further north than Rostov-on-Don) revealed that the majority of clinical manifestations of dirofilariasis occurred in the autumn months (Table 3), suggesting that the season of transmission was prolonged under the influence of climate warming. As a result, in the majority of cases, a short incubation period of 4–6 months duration could occur, and the disease could have been contracted during the same year [35].

#### 2.2.2. Urban vs. Rural Foci of Dirofilariasis

The available data indicate that the majority of registered cases of dirofilariasis are confined to urban areas, especially to large cities/administrative centers in the regions. Such variation may be attributed to the fact that dirofilariasis is still relatively new to not only the general population of Russia but also to medical personnel, particularly those at the periphery of health services. Therefore, patients might choose to travel to large cities to ensure expertise for treatment purposes. In the process of collecting the details of a case, the health treatment facility address may sometimes be recorded as the patient’s residential address. Another possibility is that the site of contraction of the infection was different from the location of the treatment facility, particularly in cases with long incubation periods.

Furthermore, the conditions for the transmission of filariasis in urban areas in general and in large cities in particular are objectively more favorable than those elsewhere. Urban areas are usually significantly warmer by up to 2–3 °C than their rural surroundings due to specific land-cover modifications and anthropogenic factors. Such a temperature anomaly is known as an urban heat island effect, which may favor the transmission of dirofilariasis [1]. The magnitude of the urban heat islands is comparable to the magnitude of the observed climate change.

Russia has faced significant changes in climate over the past decades, particularly since the 1970s [7]. For example, in the Moscow region, between 1977 and 2016, the average rate of increase in the mean summer temperature (June–August) was 0.6 °C per decade in rural areas. The urban–rural temperature contrast could reach 13 °C, while the annual mean temperature difference between the city center and rural surroundings could be approximately 2 °C. The data presented in Table 4 support these views.

However, another probable factor contributing to the increased incidence of dirofilariasis in urban areas is the ability of Culex mosquitoes (one of the vectors of dirofilariasis) to breed in multistory building cellars flooded by high-level groundwater. In addition to flooding, stray dogs might frequent such cellars. Under the influence of climate change, the microclimates in flooded cellars are favorable for the breeding of mosquitoes year-round and might result in the establishment of microfoci of dirofilariasis with perennial transmission, thus contributing to the prolongation of the local transmission of *Dirofilaria*.

#### 2.2.3. Female-Male Ratio

The incidence of dirofilariasis in women is higher than that in men. On average, the sex ratio exceeds 2 to 1. Remarkably, during the 1990s, 90% of index cases in various locations in Russia were cases in women [36]. An impact of climate change on sex differences with respect to the incidence of dirofilariasis was observed, considering the following circumstances.

Evidence exists that climate warming primarily reduces the durations of the winter and spring months [1,2], thus allowing the earlier breeding of vectors and a prolongation of the period of agricultural activity. In Russia, a considerable number of urban dwellers inhabit so-called “dachas”, which are simple country houses with small plots of attached land suitable for gardening and vegetable cultivation for self-consumption. In general, women engage in these activities much more often than men. Thus, women have a higher rate of exposure to infected mosquitoes. Dachas are also preferable places for recreational activities for other members of the family, mainly during the weekends, and contraction of infection is possible.

#### 2.2.4. Impact of Climate Warming on the Source of Infection

*Dirofilaria*-infected dogs and cats were detected in the territory of the USSR as early as the 1930s [37]. In the following decades, *Dirofilaria*-infected animals and sporadic cases of human infection were reported in territories of the Krasnodar and Astrakhan regions and in the Republic of Dagestan [38,39].

The impact of climate change on the increasing prevalence of infection in dogs was initially noted in the Rostov-on-Don region, Republic of Kalmykia and in the Volgograd region [31,32,33,40].

The data presented in Figure 5 illustrate the dynamics of the prevalence of dirofilariasis in dogs in the territory of the Rostov-on-Don region (47°23′ N) during 1997–2005 [32].

It was also observed that the infection rate in female dogs was 2.5 times higher than that in male dogs [32].

Thus, the prevalence of Dirofilaria infection in dogs under the impact of climate change increased dramatically in the territory within a relatively short period of time and was responsible for 169 human cases during the period under report [32].

The data available for the Moscow region (55°45′ N) reveal the expansion of administrative territories harboring infected dogs during 2003–2007 (Table 5).

The expansion of territories with infected dogs under the influence of climate change was accompanied by the importation of the source of infection both within and outside the country.

For example, the first case of dirofilariasis in the territory of the Volgograd region was recorded in an imported pit bull from the USA that died in 1998. The first indigenous case occurred in a local dog in 1999, which was followed by 12 cases reported in 2001, 56 cases in 2002 and 86 cases in 2004 [20].

The deployment of police service dogs in the Chechen Republic in the early 2000s played a very important role in the establishment of the local transmission of Dirofilaria in the territories under the influence of climate change. The Chechen Republic was one of a few Dirofilaria-endemic territories known since the 1970s. Service dogs from the Kirov region (58°59′ N), Novgorod region (58°26′ N), Nizhni Novgorod region (56°32′ N), Tyumen region (57°15′ N) and from a few other regions were deployed for a few months. These territories were free of local transmission of dirofilariasis prior to the arrival of the police dogs. Two to three years after the return of the dogs, which were presumably infected in the Chechen Republic, to their respective regions, the first indigenous cases of human dirofilariasis were reported. Since then, indigenous cases of human dirofilariasis have been reported regularly in these territories [25,30,35].

#### 2.2.5. Impact of Climate Warming on Vectors

Over the last ten years, the average annual temperature in the territory of the Russia increased by 0.47 °C. This was almost 2.5 times higher than the increase to the global temperature for the same period (0.47 °C versus 0.18 °C, respectively). For example, the maximum temperature recorded in the winter of 2015–2016 was 1.3 °C.

Environmental conditions have become quite favorable for mosquito vectors of dirofilariasis. Unlike many other vector-borne parasites, Dirofilaria have a great advantage in that they can be independently transmitted by three genera of mosquitoes—Aedes, Culex and Anopheles. The two former species feed during the day and at night, respectively, and have quite satisfactory compatibility with both species of Dirofilaria—*D. immitis* and *D. repens*. In the Tula region (54, 20 N), both helminth species were found in a total of 12 species of Aedes mosquitoes in equal proportions [41]. In the Rostov-on-Don region, the vector infectivity dynamics correlate strongly with infections in dogs. For example, the maximal prevalence of dirofilariasis in dogs in the Rostov-on-Don region was observed during 2000–2007 and was accompanied by an increase in mosquito infectivity from 1.0% in 2001 to 13.6% in 2005. Representatives of all three genera of mosquitoes were found to be infected with both forms of Dirofilaria (Aedes—18.2%, Culex—10.8% and Anopheles—1.3%) [31]. Climate warming facilitated increases in the densities of these three vectors in various territories of the Russia, as shown in Table 6.

## 3. Conclusions

The reviewed data convincingly indicate that the phenomenon of global warming, particularly during the last 30–40 years, has played a major role in the expansion of human dirofilariasis in the territory of the Russian Federation. At present, the disease is well established in large parts of the European and Asian territories of the country. It took only 20–25 years for the northern border of the endemic range to expand from 43° N to 58° N. With the current tempo of climate warming in the territory of the Russian Federation, there is reason to expect a further northern shift, along with an increased rate of local transmission of dirofilariasis. This is particularly likely in the absence of efficient disease prevention and control methods.

Therefore, to meet this challenge, it is necessary to establish a comprehensive epidemiological monitoring system with strong entomological and veterinary components capable of producing epidemiological information in a timely manner. The active involvement of municipalities and the community itself is also necessary. Data derived from the monitoring system should serve as a basis for the development of prevention and control activities. There is also a need to deploy modern molecular, biological, geographic and other methods within the monitoring system, which will enable public health system staff as well as personnel of the General Health system to better understand the epidemiology and pathogenesis of this disease. As the global warming process continues, prognostic maps of the disease should be developed and regularly updated. There is an urgent need to develop efficacious treatments to be used for affected people and animals.

## Figures and Tables

**Figure 1 ijerph-19-03096-f001:**
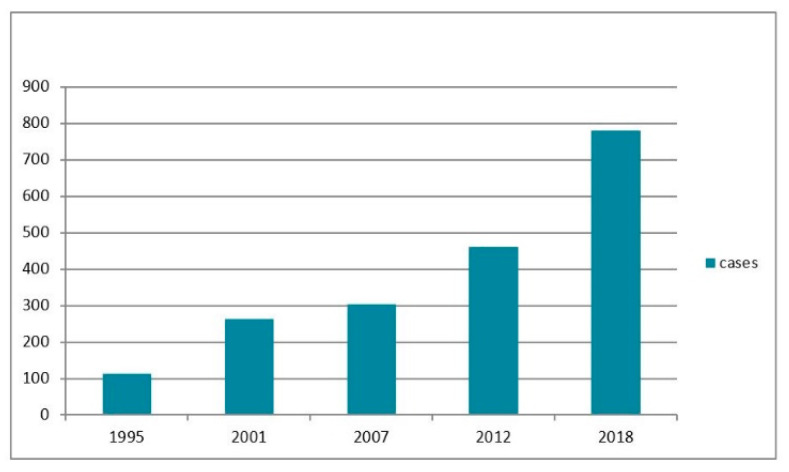
Dynamics of dirofilariasis cases, 1995–2018, Russia.

**Figure 2 ijerph-19-03096-f002:**
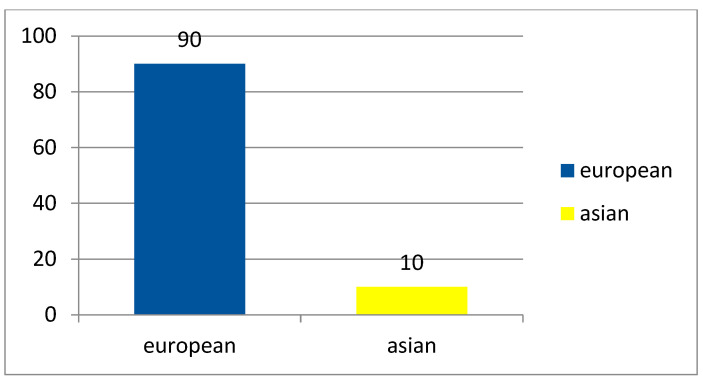
Prevalence of D. repens infections in the European and Asian parts of Russia (1997–2012).

**Figure 3 ijerph-19-03096-f003:**
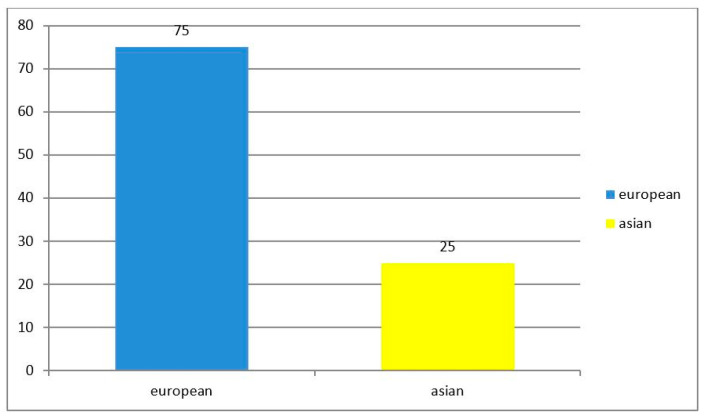
Prevalence of D. repens infection in the European and Asian parts of Russia, 2013–2018.

**Figure 4 ijerph-19-03096-f004:**
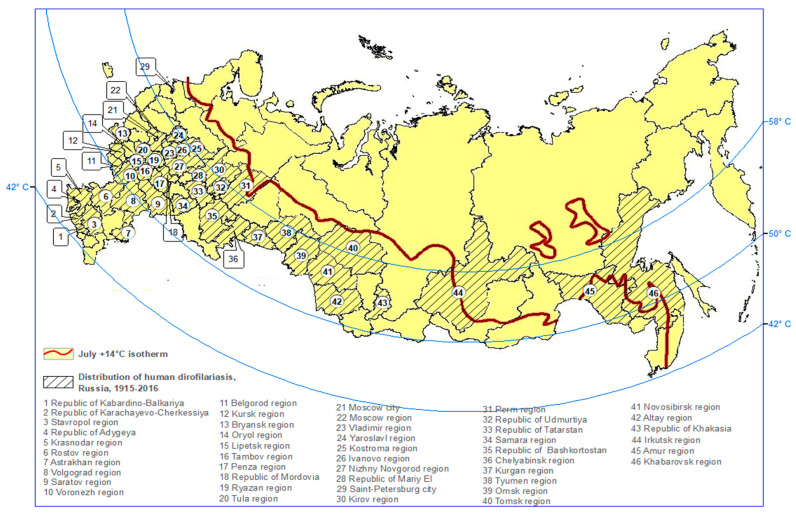
Predicted northern border of the dirofilariasis endemic area in the Russia.

**Figure 5 ijerph-19-03096-f005:**
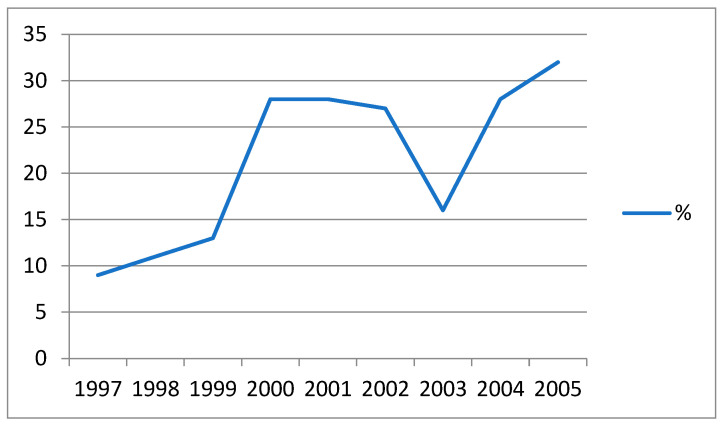
Prevalence of filariasis in dogs, Rostov-on-Don region, 1997–2005.

**Table 1 ijerph-19-03096-t001:** Dynamics of *D. repens* infections in humans in the territory of Russia (1915–2018).

Year	1915–1995 *	1996–2001	2002–2007	2008–2012	2013–2018
Cases	113	264	303	461	780

* includes cases from other ex-Union of Soviet Socialist Republics (USSR) republics.

**Table 2 ijerph-19-03096-t002:** Geographical positions of the administrative territories with local transmission of *D. repens*, Russia, 2018.

Coordinate	<45° N	46–50° N	51–55° N	56–58° N	>58° N	Total
Administrative territory	8	4	22	16	4	54

**Table 3 ijerph-19-03096-t003:** Seasonal distribution of the clinical manifestations of human dirofilariasis, Nizhny Novgorod, 2012.

Season of Clinical Manifestations	Winter	Spring	Summer	Autumn
%	19.2	24.8	23.0	33.0

Reported from [25].

**Table 4 ijerph-19-03096-t004:** Comparative incidence of human dirofilariasis in Moscow city and the Moscow region, 2013–2018.

Year/Area	2013	2014	2015	2016	2017	2018
Moscow city	28	24	16	11	9	11
Moscow region (rural)	2	3	0	0	0	0
Rest of Russia	148	130	103	70	109	89

**Table 5 ijerph-19-03096-t005:** Expansion of administrative territories (districts) harboring Dirofilaria-infected dogs, Moscow region, 2003–2007.

Year	2003	2004	2005	2006	2007
No. of Districts	19	27	30	32	33

**Table 6 ijerph-19-03096-t006:** Infection rate among Dirofilaria mosquito vectors in various territories of the Russia.

Territory	Geographic Position	Rate of Infection (%)	Source
Astrakhan region	46°34′	11.03	Kovtunov et al., 2008 [29]
Rostov region	47°23′	13.6	Nagorny et al., 2012 [31]
Tula region	54°20′	3.5	Bogacheva et al., 2016 [41]
Tomsk region	56°49′	2.3–15.8	Poltoratzkaya et al., 2018 [26]
Tyumen region	57°15′	11.7–17.9	Darchenkova et al., 2009 [17]
Novgorod region	58°26′	11.3	Rosolovski et al., 2013 [30]

## Data Availability

All documents and publications in Russian are available from the Archive and Library at the Martsinovski Institute of Sechenov University, Moscow, Russia.

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
