# Peer review of "Global Climate Change and Human Dirofilariasis in Russia"

_ijerph, 2022, doi:10.3390/ijerph19053096_

Round 1

Reviewer 1 Report

Perhaps the epidemiological situation in Russia should be discussed in connection with epidemiological data from other countries as well.

Author Response

The authors highly appreciate the valuable comments and suggestions by the Reviewers to improve the manuscript.

Comment. Perhaps the epidemiological situation in Russia should be discussed in connection with epidemiological data from other countries as well.

RESPONSE. As a matter of fact, we have already initiated writing of a special article the content of which is very well tuned to the proposal by the reviewer.

Reviewer 2 Report

The manuscript is a review of the influence of climate change on human dirofilariosis in Russia. The manuscript lacks of novelty as most information have been previously published by the same authors (Anthology of Dirofilariasis in Russia (1915-2017). Pathogens 2020 Apr 9;9(4): 275.  doi: 10.3390/pathogens9040275  doi: 10.3390/pathogens9040275).  Even accepting that the current manuscript could be an up to date of the previous one, and this should has declared in the title, the new information are very scantly and not sufficient for a full paper. Furthermore, the most of references are from Russian scientific publications, not easy to be examined by the reviewer, and this is not acceptable for an international scientific journal. As it is, the manuscript is more suitable for a national audience that for an international one. In many parts of the manuscript the authors use the term incidence where the presented data is of prevalence. It is not clear as Table 4 can support the sentence in lines 218-220.

In conclusion the current manuscript is not suitable for an international scientific journal. The authors should edit a new manuscript limiting to present the new data of human dirofilariosis in Russia.

Author Response

Comment. The manuscript is a review of the influence of climate change on human dirofilariosis in Russia. The manuscript lacks of novelty as most information have been previously published by the same authors (Anthology of Dirofilariasis in Russia (1915-2017). Pathogens 2020 Apr 9;9(4): 275. doi: 10.3390/pathogens9040275 doi: 10.3390/pathogens9040275).  Even accepting that the current manuscript could be an up to date of the previous one, and this should has declared in the title, the new information are very scantly and not sufficient for a full paper. Furthermore, the most of references are from Russian scientific publications, not easy to be examined by the reviewer, and this is not acceptable for an international scientific journal. As it is, the manuscript is more suitable for a national audience that for an international one. In many parts of the manuscript the authors use the term incidence where the presented data is of prevalence. It is not clear as Table 4 can support the sentence in lines 218-220. In conclusion the current manuscript is not suitable for an international scientific journal. The authors should edit a new manuscript limiting to present the new data of human dirofilariosis in Russia.

RESPONSE.  We have difficulties to concur with the observations by the reviewer particularly in respect of Russian published data in Russian language. The manuscript was written as an updated epidemiological data in Russia on dirofilariasis with special reference to the role of climate change.

Reviewer 3 Report

This review is specifically highlighting the correlation of climate change to spread of dirofilariasis across the Russian Federation. The review also helps to draw critical conclusions of global warming beyond Russia in countries with a similar climate. Hence, the review is worth a publication and only minor changes are recommended.

In detail:

  1. Line 86-87: explain the term effective temperature
  2. Line 133: It may be relevant to mention these are reported cases as the the monitoring of the dirofilariasis spread in Asian Russia may not be as good as in European Russia. Is anything known of dirofilariasis in wild animals such as wolves?
  3. Line 155: replace "in" with "to"
  4. Line 154: replace "upon" with "on"
  5. Line 177-180: Fuse these repetitive sentences into one only
  6. Line 210: remove ",particularly due to climate change," as cities are always warmer than the countryside independent of global climate warming
  7. Line 319: replace "necessity" with "necessary"

Take care!

Author Response

The authors highly appreciate the valuable comments and suggestions by the Reviewers to improve the manuscript.

  1. Comment. Line 86-87: explain the term effective temperature

RESPONSE. A “sum of effective temperatures” (SET) or “growing degree day” (adopted in several models in the West) needed for Dirofilaria larvae to reach infectivity. The concept of SET was originally developed and successfully applied in Russia in the control of malaria. This approach was applied to other vector-borne diseases as well, for example, for cutaneous leishmaniasis.

This sentence was entered the text of the manuscript.

  1. Comment Line 133: It may be relevant to mention these are reported cases as the the monitoring of the dirofilariasis spread in Asian Russia may not be as good as in European Russia. Is anything known of dirofilariasis in wild animals such as wolves?

RESPONSE. We accept with thanks the suggestion by the reviewer and we entered the following sentence in the text as follows  “It’s suggests that the monitoring of the dirofilariasis spread in Asian Russia may not be as good as in European Russia.”

As regards the presence of dirofilaria in wild animals, particularly in wolves, so far we did not come across with publications on the subject.

  1. Comment Line 155: replace "in" with "to"

RESPONSE. Done.

  1. Comment replace "upon" with "on"

RESPONSE. Done.

  1. Comment Fuse these repetitive sentences into one only

RESPONSE. Done.

  1. Comment Line 210: remove ", particularly due to climate change," as cities are always warmer than the countryside independent of global climate warming

RESPONSE. Done.

  1. Comment Line 319: replace "necessity" with "necessary"

RESPONSE. Done.

Round 2

Reviewer 2 Report

The authors have made a few changes to the original version. Overall, my opinion of the scientific interest of the manuscript remains the same of my previous review. The new information should have been presented in a short paper. However, if the Editor in Chief will decide to publish the manuscript as it is, I can accept this decision.